

# Response of the rare biosphere to environmental stressors in a highly diverse ecosystem (Zodletone spring, OK, USA)

Suzanne Coveley, Mostafa S. Elshahed and Noha H. Youssef

Department of Microbiology and Molecular Genetics, Oklahoma State University, Stillwater, OK, USA

## ABSTRACT

Within highly diverse ecosystems, the majority of bacterial taxa are present in low abundance as members of the rare biosphere. The rationale for the occurrence and maintenance of the rare biosphere, and the putative ecological role(s) and dynamics of its members within a specific ecosystem is currently debated. We hypothesized that in highly diverse ecosystems, a fraction of the rare biosphere acts as a backup system that readily responds to environmental disturbances. We tested this hypothesis by subjecting sediments from Zodletone spring, a sulfide- and sulfur-rich spring in Southwestern OK, to incremental levels of salinity (1, 2, 3, 4, and 10% NaCl), or temperature (28°, 30°, 32°, and 70 °C), and traced the trajectories of rare members of the community in response to these manipulations using 16S rRNA gene analysis. Our results indicate that multiple rare bacterial taxa are promoted from rare to abundant members of the community following such manipulations and that, in general, the magnitude of such recruitment is directly proportional to the severity of the applied manipulation. Rare members that are phylogenetically distinct from abundant taxa in the original sample (unique rare biosphere) played a more important role in the microbial community response to environmental disturbances, compared to rare members that are phylogenetically similar to abundant taxa in the original sample (non-unique rare biosphere). The results emphasize the dynamic nature of the rare biosphere, and highlight its complexity and non-monolithic nature.

## INTRODUCTION

Microbial communities in nature are extremely diverse. Recent high throughput 16S rRNA gene-based sequencing surveys have revealed that microbial communities in highly diverse samples exhibit a distribution pattern where the majority of bacterial species are present in extremely low numbers (*Ashby et al., 2007*; *Bowen et al., 2012*; *Logares et al., 2014*; *Pedros-Alio, 2006*; *Sogin et al., 2006*). This fraction of the microbial community has been referred to as the rare biosphere (*Sogin et al., 2006*).

Multiple studies have examined various characteristics of the rare biosphere, e.g., its proportional size within a specific sample, phylogenetic affiliations of its members,

Corresponding author
Noha H. Youssef, noha@okstate.edu

biogeography and ecology of its members, as well as its spatial and temporal dynamics in a specific habitat (*Alonso-Saez, Diaz-Perez & Moran, 2015*; *Alonso-Saez et al., 2014*; *Anderson, Sogin & Baross, 2015*; *Gobet et al., 2012*; *Hugoni et al., 2013*; *Liu et al., 2015*; *Reveillaud et al., 2014*; *Shade & Gilbert, 2015*; *Shade et al., 2014*; *Youssef, Couger & Elshahed, 2010*). All studies invariably suggest that members of the rare biosphere are phylogenetically diverse and could, collectively, mediate multiple metabolic capabilities and ecosystem functions. For example, studies have shown that members of the rare biosphere exhibit a wide range of phylogenetic novelty (*Elshahed et al., 2008*; *Lynch, Bartram & Neufeld, 2012*; *Pester et al., 2012*; *Youssef, Steidley & Elshahed, 2012*): While a fraction of the rare biosphere is typically novel (at the phylum, class, order, or family levels), many others are closely related to previously described lineages in other ecosystems. Similarly, members of the rare biosphere exhibit various levels of uniqueness (*Elshahed et al., 2008*; *Galand et al., 2009*), with a fraction being unique (i.e., bears no resemblance to other members in the community), while others are very closely related to more abundant members of the community. On a functional level, several studies have documented a large variation in the level of metabolic activity of members of the rare biosphere (by studying rRNA/rDNA transcript to gene ratios) within a single sample (*Campbell et al., 2011*), ranging from apparent dormancy to disproportionately high metabolic activity (*Besemer et al., 2012*; *Jones & Lennon, 2010*; *Logares et al., 2014*; *Pester et al., 2010*; *Pester et al., 2012*; *Wilhelm et al., 2014*).

Maintenance of members of the rare biosphere within a specific ecosystem suggests that they perform essential ecological functions. As described above, the involvement of members of the rare biosphere in specific metabolic processes within an ecosystem has been documented. Another plausible contribution of the rare biosphere to ecosystem functions is by acting as a backup system that responds to various environmental stressors e.g., temperature and pH changes, desertification, drastic change in redox potential, hydrocarbon spill (*Crump, Amaral-Zettler & Kling, 2012*; *Elshahed et al., 2008*; *Lennon & Jones, 2011*; *Marchant et al., 2002*; *Taylor et al., 2013*; *Walke et al., 2014*). Under this scenario, environmental stressors could induce the growth and promotion of specific members of the rare biosphere that are metabolically and physiologically more adapted to the new prevailing condition, and this promotion is coupled to the demotion of formerly abundant members in the community inadequately adapted to the new conditions (*Elshahed et al., 2008*). Such dynamic process contributes to the functional flexibility of a specific ecosystem. It could also explain the observed higher diversity, and functional redundancy in diverse environments, and the retention of the rare biosphere in such systems (*Shade et al., 2012*; *Yachi & Loreau, 1999*).

We reason that such hypothesis could be experimentally evaluated by stepwise subjection of a microbial community to a gradient of environmental stressors of varying magnitudes, and observing the associated patterns of promotion and demotion within rare and abundant members of the community. Here, sediments obtained from Zodletone spring, a highly diverse anoxic spring in Southwestern Oklahoma, USA, were subjected to various incremental degrees of salinity or temperature shifts. The microbial community was examined pre and post enrichment to characterize the dynamics of microbial

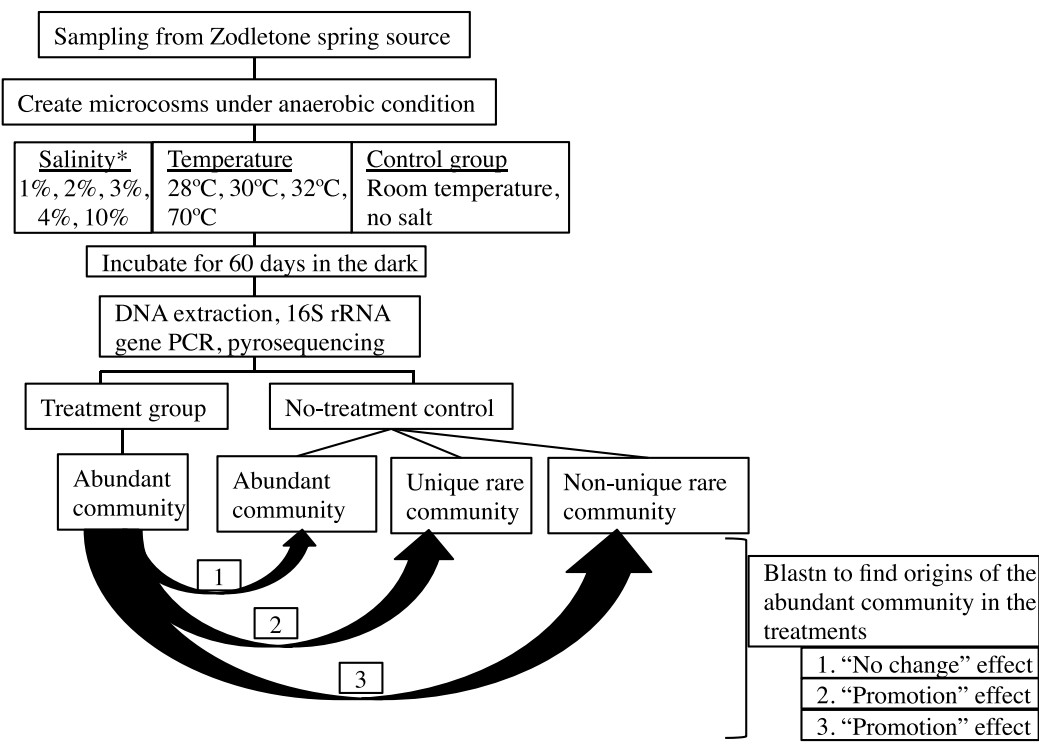

**Figure 1 A flowchart depicting the experimental design.** Abundant post-disturbance OTUs originating from an "abundant" control sequence represent the "no change" effect, where an abundant member remained abundant post-disturbance. Abundant post-disturbance OTUs originating from a "rare" control sequence represent the "promotion" effect, where a rare member of the no-stressor control was promoted to a more abundant OTU post-disturbance. This latter group could be further subdivided into two distinct categories: (a) Abundant post-disturbance OTUs "originating from non-unique rare" control sequence. And (b) Abundant post-disturbance OTUs "originating from unique rare" control sequence. * salinity percentage indicates salt concentration above ambient values (calculated at 0.9%).

community shifts associated with such processes. Our results support the proposed functionality and role of the rare biosphere in responding to environmental stressors.

## MATERIALS AND METHODS

### Sampling

Sediment samples were collected from Zodletone spring, a sulfide and sulfur-rich spring in Southwestern Oklahoma ($35°0'9''$N, $98°41'17''$W). The spring geochemistry has been previously described in detail (*Bühring et al., 2011*; *Senko et al., 2004*). The source of the spring is a contained area (1 m$^2$) with anaerobic, biomass-laden, and sulfide-rich black viscous sediments, and anoxic, sulfide-rich (8.4 mM) 40-cm water column. Spring source water was also sampled for use in enrichments as detailed below. Samples were stored on ice until returning to the lab, where they were used for the experimental procedures described below within 24 h of sampling.

### Enrichments preparation

The overall experimental setup is shown in Fig. 1.

Enrichments were prepared in Balch tubes under anaerobic conditions in an anaerobic chamber (Coy Laboratories, Grass Lake, Michigan, USA) by adding 1 g sediment to 4 ml autoclaved anoxic spring water. For this purpose, spring source water was autoclaved, cooled under a stream of $N_2$ to maintain anaerobic conditions, and used in all enrichments. To mimic environmental stress, either salinity or temperature was varied in the enrichments. Salinity was adjusted by adding NaCl to increase the concentration to 1%, 2%, 3%, 4%, or 10% above ambient concentration (measured at 0.9%) and these enrichments were incubated at room temperature. Temperature was varied by incubating enrichments at 28 °C, 30 °C, 32 °C, or 70 °C. For each condition, triplicate enrichments were incubated for 60 days in the dark. To account for any changes that could occur due to the incubation process itself, triplicate control enrichments were prepared by incubating 1g sediment in 4 ml autoclaved anoxic spring water with no further changes in salinity or temperature. This no-stressor enrichment control was subsequently used as the baseline for studying the effect of temperature and salinity change on the microbial community.

## DNA extraction, PCR amplification and sequencing

Triplicates were pooled, centrifuged at 10,000 × g for 20 min and DNA was extracted from the obtained pellet using MoBio FastDNA Spin kit for Soil (MoBio, Carlsbad, California, USA) following the manufacturer's instructions, and subsequently quantified using Qubit® fluorometer (Life Technologies, Grand Island, New York, USA). Variable regions V1 and V2 of the 16S rRNA gene were then amplified using barcoded primers for multiplex sequencing using the FLX technology. The forward primer was constructed by adding 454 Roche FLX adaptor A (GCCTCCCTCGCGCCATCAG) to the 27F primer sequence (AGAGTTTGATCCTGGCTCAG) as previously described (*Youssef, Couger & Elshahed, 2010*). The forward primer also contained a unique barcode (octamer) sequence for multiplexing. The reverse primer was constructed by adding 454 Roche FLX adaptor B (GCCTTGCCAGCCCGCTCAGT) to the 338R primer sequence (GCTGCCTCCCG-TAGGAGT). PCRs were conducted in 100 µl volume. The reaction contained 4 µl of the extracted DNA, 1 × PCR buffer (Promega, Madison, Wisconsin, USA), 2.5 mM $MgSO_4$, 0.2 mM dNTPs mixture, 0.5U of the GoTaq flexi DNA polymerase (Promega, Madison, Wisconsin, USA), and 10 µM each of the forward and the reverse primers. PCR was carried out according to the following protocol; initial denaturation at 95 °C for 5 min, followed by 35 cycles of denaturation at 95 °C for 45 s, annealing at 52 °C for 45 s, and elongation at 72 °C for 30 s. A final elongation step at 72 °C for 5 min was included. All PCR reactions were run in at least triplicates, and the resulting products of the expected size were combined and purified using QIAquick PCR cleanup kit (Qiagen Corp., Valencia, California, USA). Purified combined PCR products (11–15 µg, equivalent to ~1 µg of each enrichment condition) were sequenced using FLX technology at the Environmental Genomics Core facility at the University of South Carolina.

## Sequence quality filtering, OTU identification, and phylogenetic assignments

Sequence quality control was handled in mothur (*Schloss et al., 2009*) as described previously (*Youssef, Couger & Elshahed, 2010*). Briefly, sequences with an average quality

score below 25, sequences that did not have the exact primer sequence, sequences that contained an ambiguous base (N), sequences having a homopolymer stretch longer than 8 bases, and sequences shorter than 80 bp were considered of poor quality and removed from the data set. High-quality reads from each treatment condition were aligned against the Greengenes alignment database using a Needleman-Wunsch pairwise alignment algorithm. Filtered alignments were used to generate an uncorrected pairwise distance matrix, followed by binning the sequences into operational taxonomic units (OTUs) at 3, 6, 8, 10, and 15% cutoffs. For phylogenetic placement, representative OTUs defined at the 3% cutoff ($OTU_{0.03}$) were classified with Greengenes taxonomy scheme using the PyNAST pipeline (*Caporaso et al., 2010*). Phylum level affiliation of sequences were determined according to the classifier output, and sequences with less than 85% similarity to their closest relative in Greengenes database were considered unclassified.

## Defining the rare biosphere

The cutoff for defining rare biosphere is arbitrary and the methods used include relative abundance cutoffs, as well as frequencies of occurrence in a dataset (*Youssef, Couger & Elshahed, 2010*). For the sake of this study, we used a more relaxed definition for the rare biosphere, where rare members of the microbial community were identified using an empirical cutoff of $n \leq 10$.

## Examining microbial community response to environmental stressors

Our analysis had two main goals: To identify differences in the overall microbial community structure and diversity pre and post-enrichment, and to examine and quantify the contribution of various members of the rare biosphere in response to environmental stressors.

For the first goal (i.e., identification of differences in the overall microbial community structure and diversity due to environmental stress), microbial diversity was quantified using various diversity indices e.g., Shannon-Weiner, and Simpson diversity indices, Chao, and ACE estimators of species richness across enrichments. As well, beta diversity across samples was examined using rarefaction curve ranking since this method is not sensitive to sample size as described before (*Youssef & Elshahed, 2009*). Finally, variations in community structure due to stressors were calculated using Morisita-Horn index. This index was chosen due to its insensitivity to sample size variations (*Anderson & Millar, 2004*). Morisita-Horn index values obtained were subsequently employed to construct non-metric multidimensional scaling (NMDS) plots for community comparisons using the command nmds in mothur. The proportion of variance ($r^2$) among communities was estimated from the NMDS plots by first calculating the Euclidean distance between all pairs of data points using the equation; $d = \sqrt{(x_2 - x_1)^2 + (y_2 - y_1)^2}$, where $d$ is the Euclidean distance between 2 points of coordinates $(x_1, y_1)$ and $(x_2, y_2)$ in ordination space. The Euclidean distance was then regressed on the beta-diversity index to estimate $r^2$. In order to study the effect of the stressors on the overall phylum-level community composition, likelihood-ratio-Chi-squared test was used to examine the significant difference between
the relative abundances of phyla in the no-stressor enrichment control vs. the various enrichments.

Our strategy to achieve the second goal (i.e., understanding the contribution of the rare biosphere to environmental stressors) is based on the identification of the *abundant* members of the community post-stressor application in different enrichments, and tracing their origins to the various fractions of the community in the no-stressor control incubation. This allows for quantifying the contribution of the rare biosphere in the no-stressor control incubation to the microbial community that developed post application of stressors. The promotion of specific members of the rare biosphere in the control incubation to become abundant members in the post-stressor enrichments communities suggests a role for such rare biosphere members in responding to environmental stressors. Also, the relative abundances of such promoted members in the post-stressor enrichment community could be regarded as an indicator of the magnitude of importance of this promotion process.

Details of the analysis conducted are shown in Fig. 1. First, each sequence affiliated with the no-stressor control incubation was classified into either "abundant," "unique rare," or "non-unique rare" classes using the classification criteria detailed before (*Youssef, Couger & Elshahed, 2010*). Briefly, "abundant" refers to all sequences binned into OTUs with >10 representatives, "unique rare" refers to all sequences binned into OTUs with $\leq 10$ representatives and that are phylogenetically distinct from more abundant members of the community (>85% sequence similarity), while "non-unique rare" refers to all sequences binned into OTUs with $\leq 10$ representatives and that are phylogenetically similar to more abundant members of the community (*Youssef, Couger & Elshahed, 2010*). We then queried all abundant ($n > 10$) $OTU_{0.03}$ representatives from individual post-stressor treatments against all sequences recovered from the no-stressor control experiment (14,071 sequences) using local Blastn. Identification of the best "hit" for each abundant $OTU_{0.03}$ representative in the post-stressor incubations allowed us to trace its origin in the no-stressor control incubation into either "originating from abundant," "originating from unique rare," or "originating from non-unique rare" sequence. Abundant post-stressor OTUs originating from an "abundant" control sequence represent the "no change" effect, where an abundant member remained abundant post-stressor application. Abundant post-stressor OTUs originating from a "rare" control sequence represent the "promotion" effect, where a rare member in the no-stressor incubation control was promoted to a more abundant OTU post-stressor. This latter group could be further subdivided into two distinct categories: Abundant post-stressor OTUs "originating from non-unique rare" control sequence, and abundant post-stressor OTUs "originating from unique rare" control sequence. Based on these results, we calculated the percentage of post-stressor abundant OTUs that were "originating from abundant," "originating from unique rare," or "originating from non-unique rare" sequences, and correlated these values to the severity of enrichment (salinity or temperature) using Pearson correlations.

### Controlling the false rate of discovery using Benjamini–Hochberg adjustment

*P*-values were calculated for all Pearson correlation coefficients. Due to the large number of correlations conducted in this study, and to avoid any spurious correlations that might arise by chance (rejecting the null hypothesis due to a high p-value, when in fact the null hypothesis is true), we used the Benjamini–Hochberg procedure (*Benjamini & Hochberg, 1995*) to adjust for the false discovery rate. The command p.adjust in R was used.

## RESULTS

### Overall sequencing results

A total of 74,017 high quality sequences were obtained in this study. Estimates of diversity, species richness, evenness, and beta diversity at the species level ($OTU_{0.03}$) in various enrichments are shown in Table 1. As expected, multiple sequences affiliated with various sulfur-metabolizing lineages were identified within the datasets, including members of the sulfate- and sulfur-reducing $\delta$-Proteobacteria, chemolithotrophic sulfur-oxidizing $\beta$- and $\gamma$-Proteobacteria, as well as purple ($\alpha$- and $\gamma$-Proteobacteria) and green (Chloroflexi and Chlorobia) sulfide- and sulfur-oxidizing anoxygenic phototrophs (Table S1).

### Effect of environmental stressors on overall microbial community

In general, the increase in temperature negatively affected the community diversity (measured as rarefaction curve rank, Fig. S1A and Fig. S1B) (Pearson correlation coefficient $= -0.67$, $P = 0.17$, Fig. S2A), with the magnitude of diversity loss directly correlated to the prevailing temperature. On the other hand, all salinity treatments resulted in an increase in community diversity compared to the no-treatment control, as evident by diversity ranking estimates obtained (Table 1). However, that increase was not directly correlated to increments in salinity (Pearson correlation coefficient $= 0.06$, Fig. S2B).

At the phylum level, 50 distinct bacterial phyla and candidate phyla were identified in this study. The highest phylum level diversity was observed in sediments obtained directly from the spring prior to enrichment (50 phyla), followed by the no-treatment control enrichment (35 phyla, Table S1 and Fig. 2). Similar to diversity statistics at the putative species level, the number of phyla identified was negatively correlated to the enrichment temperature (Pearson correlation coefficient $= -0.66$, $P = 0.17$, Fig. S1A and Fig. S1B), while salinity had no clear effect on phylum level diversity (Pearson correlation coefficient $= -0.09$) (Table 1).

Likelihood-ratio-Chi-squared test was used to examine the significant difference between the relative abundances of phyla in the no-treatment enrichment control vs. the various enrichments. This method failed to identify any significant difference (likelihood ratio $\chi 2 = 99.4$, $p = 1$). Nevertheless, Pearson correlations between specific phyla relative abundance in the various enrichments, and the enrichment condition (salinity or temperature) identified the following patterns (Fig. 2 and Table S1). The increase in the enrichment temperature was positively correlated to the percentage abundance of Firmicutes, Aminicenantes (candidate division OP8), Parcubacteria

**Table 1 Overall diversity estimates in various enrichments vs. the no-stressor control incubation.**

| Treatment | Number of seqs | Alpha diversity | | | | | | Beta diversity |
|---|---|---|---|---|---|---|---|---|
| | | $OTUs_{0.03}$ | Number of phyla | % abundant | Simpson | Chao | Ace | Rarefaction curve rank[a] |
| No-stressor control 25 °C, 0.9% salt | 14,071 | 4,011 | 35 | 50.05 | 0.07 | 11,470 | 21,911 | 4, 1 |
| **Temperature enrichments** | | | | | | | | |
| 28 °C | 3,783 | 1,472 | 29 | 34.68 | 0.14 | 4,095 | 7,278 | 2 |
| 30 °C | 3,398 | 1,393 | 26 | 29.78 | 0.22 | 3,924 | 6,838 | 3 |
| 32 °C | 6,603 | 2,711 | 28 | 37.48 | 0.10 | 9,532 | 18,095 | 5 |
| 70 °C | 1,788 | 674 | 24 | 38.26 | 0.03 | 1,498 | 2,430 | 1 |
| Pearson[b] | | −0.66[*] | | −0.08 | | | | −0.67[*] |
| **Salt enrichments[c]** | | | | | | | | |
| 1% | 6,669 | 2,732 | 29 | 38.04 | 0.08 | 9,643 | 20,765 | 4 |
| 2% | 8,317 | 3,269 | 34 | 40.21 | 0.07 | 11,261 | 25,532 | 6 |
| 3% | 5,856 | 2,331 | 25 | 39.67 | 0.09 | 8,811 | 15,943 | 2 |
| 4% | 6,108 | 2,563 | 27 | 36.87 | 0.09 | 8,727 | 16,535 | 5 |
| 10% | 5,511 | 2,241 | 32 | 37.69 | 0.11 | 7,146 | 14,708 | 3 |
| Pearson[b] | | −0.09 | | −0.54 | | | | 0.06 |

Notes.

[a] Rarefaction ranks were generated by plotting the rarefaction curves of all enrichments to be compared on the same graph. Enrichments were then ranked from the least diverse (lowermost rarefaction curve, rank 1) to the most diverse (uppermost rarefaction curve, rank 5 for temperature enrichments, or 6 for salinity enrichments). Actual rarefaction curves used for ranking are shown in Fig. S1A and Fig. S1B. Two ranks are shown for the no-stressor control incubation corresponding to rarefaction curve ranks compared to the temperature (Fig. S1A), and the salinity (Fig. S1B) enrichments, respectively.

[b] Pearson correlation coefficient between the temperature (°C)/ salinity (%) in the first column and various indices in the table header. $P$-values for all correlation coefficients are shown in text.

[c] Salinity percentage indicates salt concentration above ambient values (calculated at 0.9%).

[*] Significant $p$-values following Benjamini–Hochberg adjustment for false rate of discovery.

(candidate division OD1), and Thermotogae (Pearson correlation coefficients = 0.98 ($P = 0.0005$), 0.9 ($P = 0.014$), 0.98 ($P = 0.0003$), and 0.75 ($P = 0.1$), respectively, Fig. S3A, Fig. S3B, Fig. S3C and Fig. S3D), and negatively correlated to the percentage abundances of Bacteroidetes, Marinimicrobia (candidate division SAR406), Verrucomicrobia and Latescibacteria (candidate division WS3) (Pearson correlation coefficients = −0.9 ($P = 0.016$), −0.75 ($P = 0.1$), −0.82 ($P = 0.06$), and −0.75 ($P = 0.1$), respectively, Fig. S3A, Fig. S3B, Fig. S3C and Fig. S3D). The increase in enrichment salt concentration was positively correlated to the percentage abundance of Firmicutes, and Thermotogae (Pearson correlation coefficients = 0.53 ($P = 0.258$), and 0.52 ($P = 0.269$), respectively, Fig. S3A, Fig. S3B, Fig. S3C and Fig. S3D), and negatively correlated to the percentage abundances of Verrucomicrobia and Latescibacteria (candidate division WS3) (Pearson correlation coefficients = −0.6 ($P = 0.18$), and −0.73 ($P = 0.08$), respectively, Fig. S3A, Fig. S3B, Fig. S3C and Fig. S3D). All $P$-values shown in boldface were significant following Benjamini–Hochberg adjustment (*Benjamini & Hochberg, 1995*).

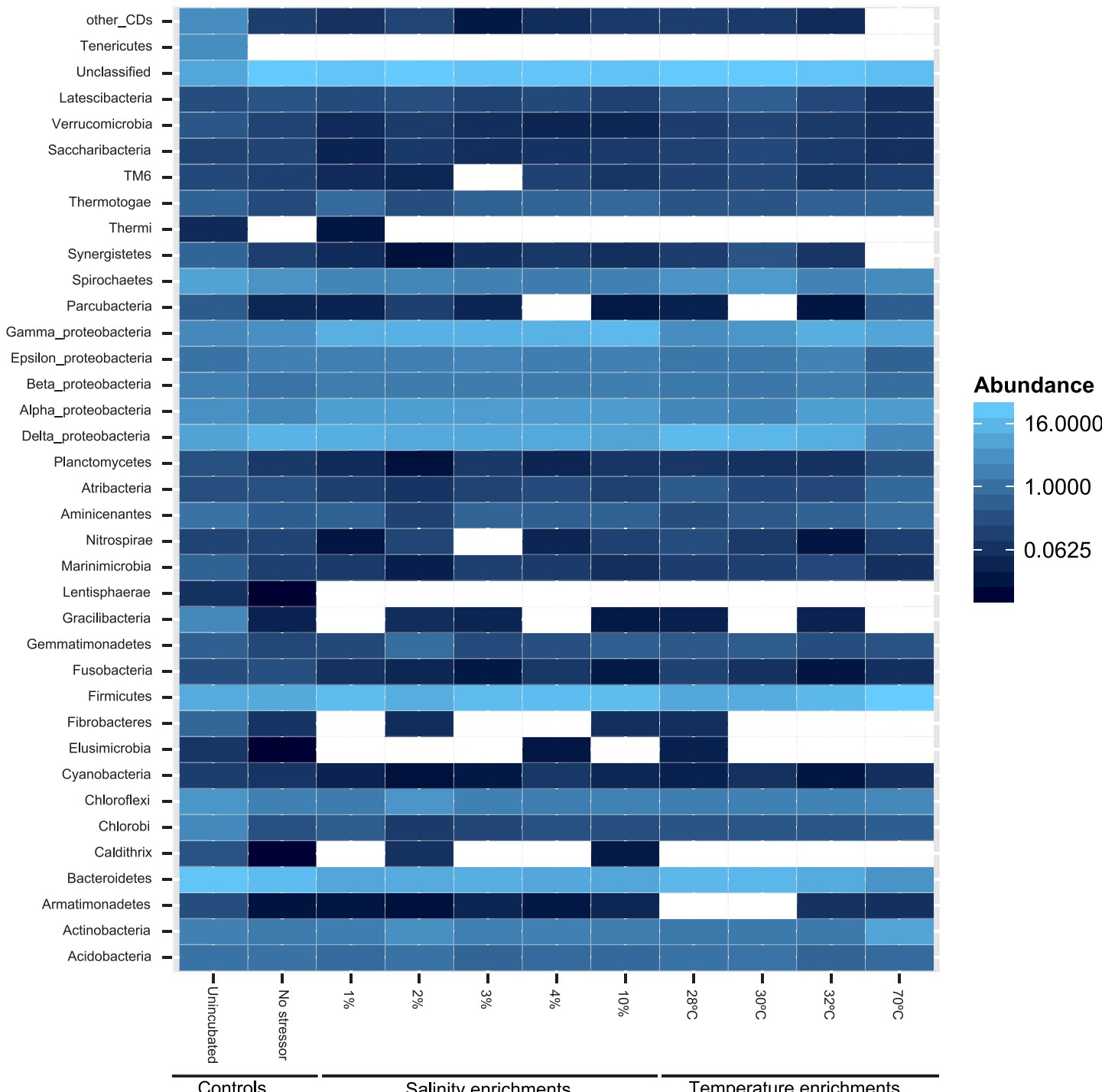

**Figure 2 Heatmap of the percentage abundance of phyla encountered in the different enrichments vs. the un-incubated and the no-stressor control.** Gracilibacteria denotes candidate division GN02, Marinimicrobia denotes candidate division SAR406, Aminicenantes denotes candidate division OP8, Atribacteria denotes candidate division OP9, Parcubacteria denotes candidate division OD1, Saccharibacteria denotes candidate division TM7, and Latescibacteria denotes candidate division WS3. Other CD denotes other candidate divisions including SPAM, AD3, CV51, KSB3, NC10, WS6, WPS-2, BRC1, SR1, WS6, ZB3, H-178, GN04, Hyd24-12, OP1, and OP11.

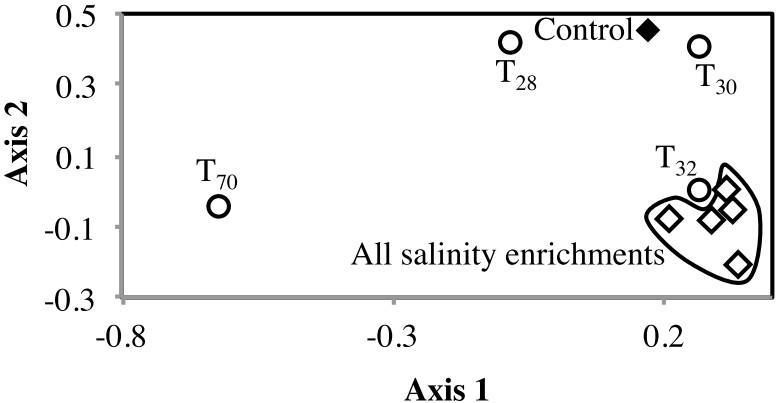

**Figure 3 Non-metric multidimensional scaling based on pairwise Morisita-Horn dissimilarity indices between abundant members of all enrichments (stress value = 0.109, the two axes explain 79.5% of the variance).** Each symbol represents one enrichment condition. The temperature enrichments abundant communities are shown by ($\bigcirc$) and each is labeled with the temperature. The salinity enrichments abundant communities are shown by ($\diamond$). The no-treatment control abundant community is shown by ($\blacklozenge$). All salinity post-disturbance abundant communities clustered together away from the no-treatment control incubation, while the effect of temperature incubations on the abundant community structure was more complex. Abundant communities following incubations at 28 °C and 30 °C had a structure more similar to the no-treatment control (which was incubated at room temperature) than the abundant community following incubation at 32 °C. The most drastic effect on the abundant community structure was the 70 °C incubation. The 70 °C abundant community clustered alone to the far left of the NMDS plot reflecting a major shift in community structure, and showed an average Morisita Horn index of $0.94 \pm 0.04$.

## Shifts in dominant microbial populations post-stressor application

To examine the impact of stressors on the microbial community, we examined the occurrence and magnitude of shifts in the structure of abundant community members post enrichments, compared to the no-treatment control. Multiple lines of evidence suggest a shift in the abundant community post-stressor application: The proportion of sequences belonging to abundant OTUs ($n > 10$) decreased post enrichment (Table 1). In addition, high levels of beta diversity were observed between the no-treatment incubation and all salinity and temperature enrichments. The abundant no-treatment community showed Morisita Horn indices of $0.63 \pm 0.07$, and $0.57 \pm 0.3$ for all possible pair-wise comparisons with the salinity and temperature post-stressor abundant communities, respectively. Indeed, non-metric multidimensional scaling plot using Morisita-Horn indices clearly shows a shift in the abundant community structure following stress application (Fig. 3). Finally, phylogenetic affiliations of abundant members of the community following application of stressors showed marked differences from those in the no treatment control (Fig. 4).

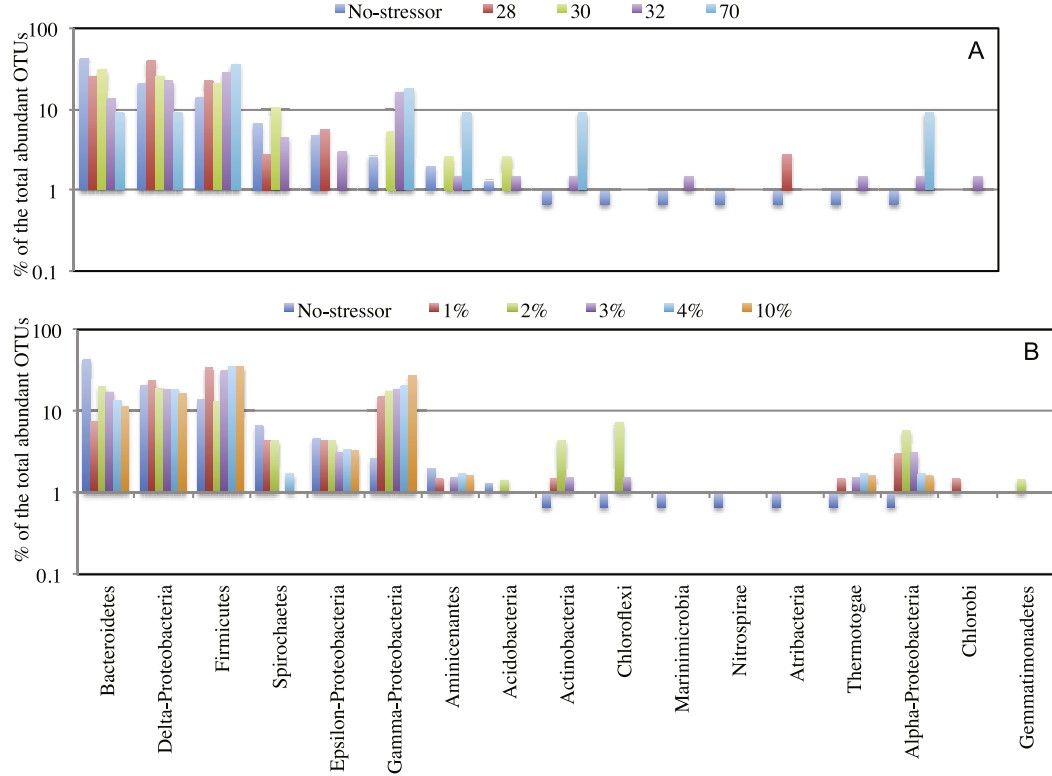

**Figure 4 Effect of salinity, and temperature on the phylogeny of abundant OTUs$_{0.03}$ following enrichment.** Effect of temperature (A), and salinity (B) on the phylogeny of abundant OTUs$_{0.03}$ following enrichment. The $Y$-axis (logarithmic scale) shows the percentage of abundant OTUs affiliated with each phylum on the $X$-axis as a fraction of all abundant OTUs encountered in the post-disturbance enrichment. At the species level (OTU$_{0.03}$), abundant OTUs ($n > 10$) in the no-treatment control belonged to the phyla Acidobacteria, Actinobacteria, Bacteroidetes, Chloroflexi, Firmicutes, Marinimicrobia (candidate division SAR406), Nitrospirae, Aminicenantes (candidate division OP8), Atribacteria (candidate division OP9), Proteobacteria, Spirochaetes, and Thermotogae. This phylogenetic profile of abundant OTUs was maintained in all enrichment conditions with the following exceptions. The increase in the salt concentration of the enrichment (A): 1. Recruited sequences belonging to Chlorobi (1% salt enrichment), and Gemmatimonadetes (2% salt enrichment) to the abundant biosphere. 2. Demoted all sequences belonging to Marinimicrobia (candidate division SAR406), Nitrospirae, and Atribacteria (candidate division OP9) to the rare biosphere. 3. Resulted in the decrease in the number of abundant OTUs belonging to Bacteroidetes, and Spirochaetes, and the increase in the number of abundant OTUs belonging to Firmicutes, Chloroflexi, Actinobacteria, and Thermotogae. On the other hand, the increase in the enrichment incubation temperature (B): 1. Recruited sequences belonging to Chlorobi (32 °C enrichment) to the abundant biosphere. 2. Demoted all sequences belonging to Chloroflexi, and Nitrospirae to the rare biosphere. 3. Resulted in the decrease in the number of abundant OTUs belonging to Bacteroidetes, and the increase in the number of abundant OTUs belonging to Firmicutes, and Actinobacteria. Also, a significant increase in the percentage of abundant OTUs belonging to Aminicenantes (candidate division OP8) was observed in the 70 °C enrichment, and a significant increase in the percentage of abundant OTUs belonging to Atribacteria (candidate division OP9) was observed in the 28 °C enrichment.

## Origins of the abundant community members following enrichment support a role for the rare biosphere in responding to environmental stressors

We hypothesized that the differences observed in dominant members of the community following application of stressors are due to recruitment of organisms from the rare

**Table 2** Tracing the origins of abundant members in elevated salinities and temperature incubations into various fractions within no-stressor control incubation.

| | % of abundant OTUs in temperature and salinity enrichments recruited from[a] | | |
|---|---|---|---|
| | Abundant | Unique rare | Non-unique rare |
| **Temperature enrichment** | | | |
| 28 °C | 91.14 | 1.77 | 7.08 |
| 30 °C | 86.96 | 6.01 | 7.03 |
| 32 °C | 58.62 | 36.93 | 4.44 |
| 70 °C | 36.7 | 60.67 | 2.63 |
| Pearson[b] | −0.87[*] | 0.86[*] | −0.86[*] |
| **Salt enrichment[c]** | | | |
| 1% | 45.44 | 44.34 | 10.21 |
| 2% | 56.17 | 37.4 | 6.43 |
| 3% | 47.31 | 46.1 | 6.58 |
| 4% | 47.78 | 44.76 | 7.46 |
| 10% | 41.98 | 54.55 | 3.47 |
| Pearson[b] | −0.6[*] | 0.85[*] | −0.86[*] |

**Notes.**

[a] The percentage of abundant OTUs in the temperature and salt enrichments that had Blastn best "hits" in the no-treatment control belonging to abundant, unique, and non-unique rare fractions. The percentage of abundant OTUs that were recruited from the unique rare fraction of the no-treatment control also includes those abundant OTUs that had no hits in the no-treatment control using the criteria described in 'Materials and Methods'.

[b] Pearson correlation coefficient between the temperature (°C)/ salinity (%) in the first column and percentages recruited from the different fractions in the table header. $P$-values for all correlation coefficients are shown in text.

[c] Salinity percentage indicates salt concentration above ambient values (calculated at 0.9%).

[*] Significant $p$-values following Benjamini–Hochberg adjustment for false rate of discovery.

biosphere to become part of the abundant members of the communities. To this end, we sought to identify the origin of all members of the abundant community post-stressor application (Fig. 1) and determine their origin (abundant, rare unique, and rare non-unique) in the no-treatment control community. Our analysis identified three distinct patterns (Table 2): First, post-enrichment abundant OTUs that were similar to "abundant" sequences in the no-treatment control. This group constituted 36.7–91.1% of the community in various enrichments. As expected from the community structure analysis, the highest percentages were encountered with the 28 °C and 30 °C incubations, since the abundant community from these temperature incubations was very similar to the no-treatment control (Fig. 3). The percentage of this group decreased as the enrichment salinity, and temperature increased (Pearson correlation coefficient = −0.6, and −0.86 ($P = 0.07$) for salinity, and temperature, respectively, Fig. S4A and Fig. S4B). The second group constitutes abundant OTUs in the enrichments (8.9–63.3%) that were recruited from the no-treatment control rare biosphere, providing direct evidence that the rare biosphere could act as a backup system to respond to environmental stressors. Within this group, we differentiate between two distinct fractions: (A) Post-enrichment abundant OTUs that were promoted from "rare non-unique" sequences in the no-treatment control, i.e., rare members of the original community phylogenetically similar to more

abundant members of the community. Of the total number of abundant OTUs, this fraction constituted 2.6–10.2% in various conditions. This percentage decreased as the enrichment salinity, and temperature increased (Pearson correlation coefficient $= -0.86$ for both salinity ($P = 0.035$), and temperature ($P = 0.076$), Fig. S4A and Fig. S4B). And (B) Post-enrichment abundant OTUs that were promoted from "rare unique" sequences in the no-treatment control, i.e., rare members of the original community phylogenetically distinct from more abundant members of the community. Of the total number of abundant OTUs, this fraction constituted 1.8–60.7% of abundant OTUs in various conditions. This percentage increased as the enrichment salinity, and temperature increased (Pearson correlation coefficient $= 0.85$ ($P = 0.039$), and $0.86$ ($P = 0.072$) for salinity, and temperature, respectively Fig. S4A and Fig. S4B). Therefore, we argue that, while both factions of the rare biosphere are important in responding to changes in environmental conditions and both act to recruit members to the abundant community, the magnitude of contribution of the "non-unique" rare biosphere to the promotion process was less significant. On the other hand, the "unique" rare biosphere seemed to contribute to the promotion process both when the magnitude of environmental stressors applied was slight, e.g., similar to what would happen during diurnal variation in salinity and temperature, as well as severe, e.g., similar to what would be encountered during seasonal variation in temperature or salinity, or following a drastic change in environmental condition, e.g., drought or fire. This latter effect is so evident in the 10% salt, and the 70 °C incubation, where 55%, and 61%, respectively of the abundant community was recruited from the "unique" rare biosphere (Table 2).

## DISCUSSION

In this study, we experimentally evaluated the response of microbial community from Zodletone spring source sediments to environmental stressors (various levels of elevated salinities or temperatures), with emphasis on understanding the role of the rare biosphere in the process. We demonstrate that rare bacterial taxa could be promoted to abundant members of the community following environmental manipulations. The magnitude of this promotion process is directly proportional to the severity of the stress applied. Finally, rare members that are phylogenetically distinct from abundant taxa in the original sample (unique rare biosphere) play a more important role in microbial community response to environmental stressors.

The results are in agreement with several previously proposed ideas regarding the function and maintenance of the rare biosphere. Prior studies have speculated that the rare biosphere acts as a backup system that preserves microbial community-level function in face of stress (*Elshahed et al., 2008*; *Lynch & Neufeld, 2015*; *Reid & Buckley, 2011*). This fraction of the rare biosphere has been referred to as "conditionally rare taxa". The occurrence of conditionally rare taxa in multiple environments has recently been documented (*Aanderud et al., 2015*; *Shade & Gilbert, 2015*; *Shade et al., 2014*; *Sjostedt et al., 2012*; *Taylor et al., 2013*; *Walke et al., 2014*). These taxa show cyclical low abundance until conditions become favorable, where they respond to the change and increase in abundance. Previous

studies showed that such taxa could make up to 28% of the community (*Shade et al., 2014*). Conditionally rare taxa contribute greatly to the diversity of an ecosystem. The presence of a diverse low-abundance fraction increases the ecosystem's ability to maintain its functions during environmental changes (*Shade et al., 2012*; *Yachi & Loreau, 1999*). In the current study, rare taxa in the no-treatment control incubation that were promoted to become abundant members in different incubations constituted 8.9–63.3% of the microbial community under various conditions of elevated temperatures or salinity. This provides direct evidence that this fraction in the rare biosphere acts as a backup system.

Within these conditionally rare taxa that increased in abundance in response to environmental stress, we differentiate between two different fractions, those that were promoted from "rare unique" sequences in the no-treatment control, i.e., rare members of the original community phylogenetically distinct from more abundant members of the community, and those that were promoted from "rare non-unique" sequences in the no-treatment control i.e., rare members of the original community phylogenetically similar to more abundant members within the original community. Our results suggest that the rare unique members of the microbial community actively contributes to the ecosystem's response to stress, since the majority of conditionally rare taxa identified in various enrichments were unique in the no-treatment control incubation, e.g., 37% to ~61% for temperature incubations ≥32 °C, and 37% to ~55% for salt incubations (Table 2). The magnitude of contribution of the rare unique fraction to the abundant community increased with the severity of stress (i.e., with the increase in salinity as well as temperature of enrichment). Interestingly, the abundant community in the most severe "unnatural" condition (the enrichment at 70 °C) was mostly (60.7%) made-up from representatives of the "unique" rare biosphere (Table 2), confirming what was shown before in arctic marine sediments (*Hubert et al., 2009*), and freshwater stratified lakes (*Shade et al., 2012*) that a fraction of the conditionally rare taxa exhibit greatly reduced metabolic activity under the natural environmental conditions but is able to exploit the "forced" manipulation of conditions (e.g., high temperature incubations of arctic sediments, or complete anoxic conditions in the hypolimnion layers of the lake) and become abundant. The "non-unique" rare biosphere, on the other hand, did not significantly contribute to the abundant community post-stressors, with the highest representation of its members to the post-stress abundant community being 10% in the 1% salt enrichment (Table 2), and the magnitude of its contribution to the abundant community decreasing with the severity of stress.

In addition to conditionally rare taxa, the post-stress abundant community was also partly made-up (36.7–91.1%) of previously abundant members in the no-treatment control, where microorganisms capable of coping with the new condition (increased salt or temperature) remained abundant. It is telling that the highest magnitude of contribution of the pre-stress abundant members to the post-stress abundant community was observed in the 28 °C (91.1%) and 30 °C (85.6%) enrichments, since these two conditions are the closest to the no-treatment control. While those numbers could possibly be inflated by lingering DNA from cells that were originally abundant in the no-treatment control but

that lysed or became inactive during enrichment (what has previously been referred to as taphonomic gradient), the observation that the magnitude of contribution of the pre-stress abundant members to the post-stress abundant community decreased with the increase in the severity of stress is expected. This is due to the fact that abundant members of the community under certain conditions are those microorganisms that are most adapted to their current environment, and as the conditions change, those members are also expected to change in abundance and/or metabolic activity. That also explains the difference observed in the abundant community structure pre and post application of stress (Fig. 4).

## CONCLUSIONS

In conclusion, the current study provided direct evidence for the contribution of members of the rare biosphere to the abundant community post environmental stress, and hence confirming the notion that a fraction of the rare biosphere acts as a backup system that readily responds to environmental stressors. While we show here, similar to previous studies (*Lynch & Neufeld, 2015*, and references within), that the rare biosphere responds to both periodical (e.g., temperature and seasonal changes), as well as drastic (e.g., what would be encountered in drought or fire) changes in the ecosystem conditions, the disproportionate contribution of rare members whose phylogenetic affiliations are distinct when compared to more abundant members of the community has not been shown before and reinforces the phylogenetic and metabolic complexity and diversity of the rare biosphere.

### Funding
This work was supported by the National Science Foundation Microbial Observatories Program (Grant EF0801858). The funders had no role in study design, data collection and analysis, decision to publish, or preparation of the manuscript.

### Grant Disclosures
The following grant information was disclosed by the authors:
National Science Foundation Microbial Observatories Program: EF0801858.

### Competing Interests
The authors declare there are no competing interests.

### Author Contributions
- Suzanne Coveley performed the experiments, analyzed the data, wrote the paper, prepared figures and/or tables, reviewed drafts of the paper.
- Mostafa S. Elshahed conceived and designed the experiments, analyzed the data, contributed reagents/materials/analysis tools, wrote the paper, prepared figures and/or tables, reviewed drafts of the paper.

- Noha H. Youssef conceived and designed the experiments, performed the experiments, analyzed the data, contributed reagents/materials/analysis tools, wrote the paper, prepared figures and/or tables, reviewed drafts of the paper.

## DNA Deposition

The following information was supplied regarding the deposition of DNA sequences:

Datasets can be found at MG-RAST (https://metagenomics.anl.gov/). The Metagenome IDs associated with the datasets are: 4644179.3, 4644181.3, 4644180.3, 4644182.3, 4644183.3, 4644184.3, 4644185.3, 4644186.3, 4644187.3, 4644188.3.

## Supplemental Information

Supplemental information for this article can be found online at http://dx.doi.org/10.7717/peerj.1182#supplemental-information.

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
