# Peer review of "Response of the rare biosphere to environmental stressors in a highly diverse ecosystem (Zodletone spring, OK, USA)"

_PeerJ, doi:10.7717/peerj.1182_

## Round 0.1 · original submission · Major Revisions

Both Reviewers agree on the same issues that should be addressed. In particular the authors make a lot of inferences in their study based on Pearson Correlations. They report the correlation coefficients. However, they never show the correlation plots. I think the authors should present this information, if not in the main manuscript, then as Supplementary Materials. Some of the r-values are high, but it would be nice to see if the data meet the assumption of normality and equal variance. Also, the authors need to provide p-values associated with the correlation analyses. Also It is surprisingly that you don't detect sulfate-reducing bacteria in the analysis, or any related taxa capable of metabolizing sulfur compounds. Why in a sulfur rich sediment there are not "sulfur" representatives?. This is other reason why you need to know the "real" community structure prior incubation.

In the introduction it is mention the plausible contribution of the rare biosphere to the ecosystem functions by acting as a backup or seed system that responds to various levels of environmental fluctuations. However it is not discussed the plausible functional roll of the OTUs found.

·

Basic reporting

In general, the manuscript is reasonably clear and well-written. I do have some questions and suggestions regarding the basic reporting.

1) There are some colloquialisms and awkward phrases that I recommend be removed or fixed, such as "bewildering array" (line 32); "monolithic groups" (line 50 and elsewhere); "to zoom in on the impact" (line 248); "multiple evidences" (line 251); "the results provide experimental proof" (line 310); "wide range of uniqueness" (line 55); "Evidences of the presence of" (line 314); what is meant by "keystone physiological functions" (line 65); "severity of disturbance is highly logical" (line 363) based on what theory?

2) There are some ecological and evolutionary concepts that should be clarified or corrected:

a) It is very important not to equate the rare biosphere with seed banks ("seed system" is an awkward phrase not commonly used). In ecology, seed banks refer to a collection of inactive or dormant individuals, which may be rare or abundant.

b) Disturbances refer to the removal or loss of individuals from a system. It would be more appropriate from an ecological perspective to refer to the temperature and salinity shifts as "stressors".

c) The authors use the term resiliency in many places. There is a very specific definition for resilience (the rate at which a system recovers following a perturbation). The data in this manuscript would not allow one to quantify resilience.

d) The authors make reference to "fitness" (line 355). It's not clear that the authors can make any inferences about the fitness of taxa from the data they've collected.

e) On line 64 the authors seem claim that the existence of rare taxa suggests the play unique functions. It's important to note that hollow-shaped rank abundance curves can easily be generated from random sampling (neutral processes), where no unique traits or properties are associated with a taxon. So, the logic of this sentence is not entirely sound.

f) On line 320, the authors state "α-diversity of an ecosystem"; alpha-diversity can only be measured for a sample.

g) On line 322, Can the results from this study with a single time point be used to make inferences about the taxa being conditionally rare? Or are more time points needed?

3) In the introduction, I didn't find the first paragraph to be very effective because it emphasizes sequencing technology. In my opinion, this front-loading of methods (454 sequencing in this study) detracts from the more interesting questions being addressed.

4) There are certain stylistic issues that should be addressed. For example, starting on line 272 and in other places, the authors have a tendency to present there ideas as lists (e.g., ".2. And") in a way that was awkward and hard to follow. In many places (e.g., starting on line 280), there are many vague references ("those" and "This"), which made arguments unclear. Last, there are many places with grammatical errors, including verb/subject agreements ("High levels of beta diversity was observed" [line 252]); the manuscript should be carefully edited to fix these issues.

5) Is it critical to have "(Zodletone spring, OK, USA)" in the title?

Experimental design

1) On line 96, the authors state that "Samples were stored on ice until returning to the lab" until the experiments were initiated, which was less than 24 hrs. This seems somewhat undesirable to store samples on ice (below freezing) and then subject some samples to 70C temperatures. What was the temperature of the samples when taken from the field?

2) Perhaps this is more of an Introduction comment, but no real justification is given for why samples were challenged with temperature and salinity stress. What's the context?

3) On line107, the authors explain that they went to effort to run each treatment combination in triplicate. This is good. However, on line 111, it sounds as though the extractions from the experimental units within a treatment were then pooled. If this is true, it's very unfortunate. The authors may want to check out this paper, which addresses the pooling issue: http://www.ncbi.nlm.nih.gov/pubmed/20438583

4) Line 141: It's not clear why the authors describe the cutoffs (which is sort of confusing; percent dissimilarity) since I did not take note of this aspect of diversity being revisited later in the manuscript.

5) Line 150: It's somewhat uncommon to define rarity based on total number of reads. This can only be justifiable if there are an equal number of reads across all samples. Was this definition of rarity performed on rarified data? Also, it would be useful to know what relative abundances corresponded to the n < 10 cutoff.

6) Line 158: The authors state that they quantified beta-diversity with rarefacton curves. This is strange to me, since rarefaction is most commonly used to estimate species richness, which is an alpha-diversity metric. In general, I was a bit confused by the section starting on line 151.

7) Line 196: What is meant by "original enrichment"?

8) Line 203: What about a "demotion" category? Wouldn't it be interesting to know what species dropped in the rank abundance curve?

9) Is there a justification for why the authors spend some much time focusing on phylum-level taxonomy?

10) On line 231, the author discuss likelihood-ratio-Chi-squared. This should be moved to the Methods section and the authors should more clearly describe what hypothesis this technique is being used to test.

11) Line 250 "Abundant community": does this imply that the rare taxa were not included in the analyses?

12) Figure 3: shouldn't there be stress values reported for NMDS? Also, typos/translation errors in caption.

13) Figure 4: What is "RT". Label the horizontal line at 10 for clarity. I'm not entirely sure what to take away from this figure.

14) Table 1: what is the "rarefaction curve rank"?

Validity of the findings

1) The authors make a lot of inferences in their study based on Pearson Correlations. They report the correlation coefficients. However, they never show the correlation plots. I think the authors should present this information, if not in the main manuscript, then as Supplementary Materials. Some of the r-values are high, but it would be nice to see if the data meet the assumption of normality and equal variance. Also, the authors need to provide p-values associated with the correlation analyses.

2) The description of in the Results starting on line 261 needs to be revised for clarity.

3) On line 306, the authors state "The magnitude of this promotion process is directly proportional to the severity of the disturbances". I believe this brings up the comment I had on Perason Correlations described above. The authors seem to present some of the data in questions in Figure 2, but I do not feel that this is the most convincing use of the data to support their claim. I had a similar concern about the text on line 339.

Reviewer 2 ·

Basic reporting

The main idea described in the introduction is good (rare biosphere) however, does not justify the approach used. There is not sufficient background to demonstrate the following: How the Idea of using " sulfur-rich sediments with changes in temperature and salinity" it will work to test the hypothesis
The figures and tables are not very explicit.

Figure 1. Is very confusing, a better representation of the method , should be presented.
Figure 2. The abundant OTU, could be better explained as figure 2, but with normalized data with hierarchical clustering. (see the example: http://www.nature.com/srep/2015/150130/srep08136/fig_tab/srep08136_F5.html)
Table 1 . The overall diversity estimates are very informative, however, a very explicit way to shown the ecological diversity is using rarefaction curves: See example in http://mbio.asm.org/content/3/5/e00338-12/F2.expansion.html
Table 2. The table is confusing, the abundant, unique rare and non-unique rare abundances could be presented in a bar plot.

Experimental design

The overall experimental design have a lot of limitations, shortcomings and It is not suitable for testing the hypothesis described in the introduction.
Choosing the sample:
1. There are previous studies that analyze the bacterial diversity of the Zodletone spring sediments?, It is expected a high diversity? Or should be present the main metabolic sulfur guilds: sulfate-reducing bacteria (i.e Desulfovibrio) and sulfur oxidizing bacteria (i.e chemolithotrophic sulfur-oxidizing bacteria: Beggiatoa, or anoxigenic purple or green sulfur bacteria)
If there is not sufficient information about the above, then tittle is not making sense and is inadequate
Physicochemical variables of the original sample
Which are the physicochemical variables of the original samples (i.e temperature, pH, oxygen concentration etc). Did you replicate this variables in the mesocosms experiment?
Experimental controls:
2. The "negative control" used, is not really a good control due to changes in community structure during the incubation time (60 day).
**The ideal experiment it would take into account the DNA extraction prior incubation period, in order to have the "real" community structure represented. **
3. Due to the above the method is not suitable for testing the rare biosphere hyphothesis , if yo don't know the abundances of the members of the community of the REAL community (The fact of putting the sediments in mesocosms involves a disturbance perse) .
4. Which criteria were used to decide that changes in salinity and temperature were suitable to test the hypothesis? As you described in the introduction there are other environmental disturbances ranging to pH, light exposure, nutrient levels etc.
Why salinity and temperature in those ranges? They were chosen arbitrarily? . Other environmental disturbance within the ecological context of anoxic-sediments,
it would have been the changes in sulfur (i.e Sulfate: SO42- and Sulfide H2s) concentration in the mesocosms.

Validity of the findings

1. It is surprisingly that you don't detect sulfate-reducing bacteria in the analysis, or any related taxa capable of metabolizing sulfur compounds. Why in a sulfur rich sediment there are not "sulfur" representatives?. This is other reason why you need to know the "real" community structure prior incubation.
In the introduction it is mention the plausible contribution of the rare biosphere to the ecosystem functions by acting as a backup or seed system that responds to various levels of environmental fluctuations. However it is not discussed the plausible functional roll of the OTUs found.
Which is the putative role of the abundant taxa "post-disturbance"?, the "abundant" , "Unique rare " and " " Non-unique rare" OTUS?, Introduction has a lot of content for open discussion however it's not showing it.

Additional comments

Reviews by number of line:

22: Its not clear what did you mean with non monolitic-nature. Use another sentence.
50: Did you mean “non -monophyletic group”?
55-62: All this lines could be used in the discussion.
71-77: Same as above
82: There is not sufficient backgroud to link the information with : sediments of the anoxic spring.
92: Any coordinates?
100: Salinity of the original sample (ppm?)
106: Temperature of the original sample (°C)
121: DNA concentration ng/uL
192-197: Redundant text

---

## Round 0.2 · Minor Revisions

The manuscript is greatly improved, but perhaps the authors may want the opportunity to check their stats once more before formal acceptance. The authors should be reminded that spurious correlations can arise when too many correlations are made without a proper hypothesis, one of the reviewers has a suggestion to check this carefully.

·

Basic reporting

The authors have done a good job addressing the reviewers's concerns.

Experimental design

I their rebuttal document, the authors clarify that they pooled their DNA. This should preclude some statistical analyses, but it's entirely clear to me whether or not this is a problem based on the revised manuscript.

Also, I notice that there were a lot of correlations that were conducted. This is something that happens when dealing with diverse data sets. However, the authors should be reminded that spurious correlations can arise. I would recommend performing a test for false discovery. This easy tool may help: https://stat.ethz.ch/R-manual/R-devel/library/stats/html/p.adjust.html. The Benjamini-Hochberg approach is much more forgiving than the more traditional Bonferroni.

Validity of the findings

No comments

Additional comments

See above

Reviewer 2 ·

Basic reporting

All suggested changes were properly made.
However, due to the change of title , only two references are not correct
1) The first sheet of the Reviewing Manuscript refers to the unchanged title
2)The Supplementary document- revised.docx refers to the unchanged title

Experimental design

All suggested changes were properly made.

Validity of the findings

All suggested changes were properly made.

Additional comments

All suggested changes were properly made.

---

## Round 0.3 · accepted · Accept

The manuscript is now fit for publication, thank you for all the work.